# Kinetics and mechanical work done to move the body centre of mass along a curve

**Raphael M. Mesquita**[ID], **Patrick A. Willems, Arthur H. Dewolf**[ID]\*[◐], **Giovanna Catavitello**[¤][◐]

Laboratory of Biomechanics and Physiology of Locomotion, Institute of NeuroScience, Université catholique de Louvain, Louvain-la-Neuve, Belgium

◐ These authors contributed equally to this work.
¤ Current address: Business and Decision Life Science Medical Communication Service Center, Brussels, Belgium
\* arthur.dewolf@uclouvain.be

**Data Availability Statement:** DOI 10.17605/OSF. IO/JGSH9

## Abstract

When running on a curve, the lower limbs interact with the ground to redirect the trajectory of the centre of mass of the body (CoM). The goal of this paper is to understand how the trajectory of the CoM and the work done to maintain its movements relative to the surroundings ($W_{com}$) are modified as a function of running speed and radius of curvature. Eleven participants ran at different speeds on a straight line and on circular curves with a 6 m and 18 m curvature. The trajectory of the CoM and $W_{com}$ were calculated using force-platforms measuring the ground reaction forces and infrared cameras recording the movements of the pelvis. To follow a circular path, runners overcompensate the rotation of their trajectory during contact phases. The deviation from the circular path increases when the radius of curvature decreases and speed increases. Interestingly, an asymmetry between the inner and outer lower limbs emerges as speed increases. The method to evaluate $W_{com}$ on a straight-line was adapted using a referential that rotates at heel strike and remains fixed during the whole step cycle. In an 18 m radius curve and at low speeds on a 6 m radius, $W_{com}$ changes little compared to a straight-line run. Whereas at 6 m s$^{-1}$ on a 6 m radius, $W_{com}$ increases by ~25%, due to an augmentation in the work to move the CoM laterally. Understanding these adaptations provides valuable insight for sports sciences, aiding in optimizing training and performance in sports with multidirectional movements.

## Introduction

In the laboratory, human locomotion is mostly analysed in a "steady state" condition *i.e.*, while moving on a straight line, at an average constant speed, on a level and firm terrain [1–4]. In nature [5–7], in daily life or during sport activities [8–10], this steady state situation appears rather infrequently. External factors and/or circumstances often require modifying the gait pattern to accelerate/decelerate the body [11–13], to move on a soft yielding ground [14,15], to move upwards or downwards [16,17], to leap over an obstacle [18], *etc.* Most studies on human locomotion are done in a straight line, whereas daily, one rarely moves in only one direction.

**Funding:** This study was funded by the National Fund for Scientific Research (F.N.R.S - CDR 40013847).

**Competing interests:** The authors have declared that no competing interests exist.

**Abbreviations:** $a_x$, $a_y$, $a_z$, Lateral, fore-aft and vertical component of the acceleration of the CoM in the o-x-y-z reference frame; BW, Body weight; CoM, Body centre of mass; $c_x$, $c_y$, $c_z$, Lateral, fore aft and vertical constants when integrating the acceleration to get the velocity; δ, Angle between the vector $v_h$ and the y-axis; $\delta_{TD}$, $\delta_{TO}$, Angle between the vector vh and the y-axis respectively at *TD* and *TO*; $E_{com}$, Total energy of the CoM due to its movements relative to the surroundings; $E_X$, $E_Y$, Kinetic energy of the CoM due to its velocity along the X- and the Y-axis; $E_z$, Kinetic and potential energy of the CoM due to its vertical movements; $F_x$, $F_y$, $F_z$, Lateral, fore aft and vertical components of the GRF in the o-x-y-z reference frame; $F_x$, $F_y$, $F_z$, Lateral, fore aft and vertical components of the GRF in the o-x-y-z reference frame; $F_X$, $F_Y$, $F_z$, Lateral, fore aft and vertical components of the GRF in the O-X-Y-z reference frame; GRF, Ground reaction force; m, Body mass; MTU, Muscle tendon unit; o-x-y-z, Inertial reference frame fixed to the laboratory. During straight-line running, $x$ points in the lateral direction, $y$ in the fore aft direction and the $z$-axis is vertical; O-X-Y-z, Inertial reference frame that changes at each foot contact: $O$ is the position of the foot at the instant *TD*, the $Y$-axis point in the direction of $v_h$ at *TD*, the $X$-axis is perpendicular to the $Y$-axis and the $z$-axis is vertical; $PL_{cx}$, $PL_{cy}$, Coordinates in the transverse plane of the point $PL_c$ located inside the pelvis i.e., at the centroid of the triangle <$SAT_L$, $SAT_R$, S2>; $R_{th}$, Theoretical radius determined by the running lanes; R, Actual radius of curvature calculated by a circular fit on the ($PL_{cx}$, $PL_{cy}$) coordinates during the strides analysed; r, Instantaneous radius of curvature i.e., actual distance between the points ($x_o$, $y_o$) and ($PL_{cx}$, $PL_{cy}$) at each instant of the stride; %Recovery, Percentage of energy recovered through the transduction between $E_X$, $E_Y$ and $E_z$; $SAT_L$, $SAT_R$, S2, Reflective markers placed respectively at the left and right anterior superior iliac tuberosity and at the level of the second sacral vertebra; $S_z$, Vertical displacement of the CoM; SL, Straight-line run; θ, Angle between the tangent to the circle and the y-axis; $\theta_{TD}$, $\theta_{TO}$, Angle between the tangent to the circle and the y-axis respectively at *TD* and *TO*; $t_c$ and $t_a$, Duration of the phases of contact with the ground and the aerial phases; TD, Touch down i.e., instant when the foot touches the ground; TO, Take off i.e., instant when the foot leaves the ground; $v_x$, $v_y$, $v_z$, lateral, fore aft and

When running, one can change direction in different ways [19]. For example, a runner can gradually change the direction of progression, step by step, *e.g.*, when running on an athletic track (curved running) [20–23]. Alternatively, the runner can change direction abruptly during a transition phase (lasting 1–3 steps), as in various multidirectional team sports (cutting manoeuvres [19,24–26]). Although some principles and specific hypotheses have been investigated for various modes of locomotion while changing direction [19,20,27–33] the picture is not yet complete regarding the biomechanics of curved running in humans.

To the best of our knowledge, the trajectory and potential adaptions of the centre of mass of the body (CoM) when running along a curve and the work done by the muscle-tendon unit (MTU) to sustain its movements ($W_{com}$) have not yet been analysed. Indeed, since wind resistance is negligible and the foot is not skidding on the ground [14,34], essentially all the work done by the MTU relative to the CoM (classically called $W_{ext} = W_{com} + W_{env}$ [34,35]) is done to sustain its movements ($W_{com}$) rather than to move it against the environment ($W_{env}$), therefore $W_{com} = W_{ext}$.

A further understanding of the constraints and adaptions of the CoM mechanics while running on a curve may have practical implications in various fields, including sports performance, injury prevention, and rehabilitation [35]. To date, regarding kinetic variables, only the description of the ground reaction forces during curved running has been documented in the literature, most of which has focused on sprint mechanics performed at maximal velocity on an athletic track curve where the minimal radius of curvature for an outdoor track is of 36.5 m [20–23,28,36,37] or when running in curves with radii smaller than 6 m [27].

When running in a curved path, the velocity of the CoM must be deflected each step in the transverse plane [28] and a net lateral force, opposite to the side one wishes to move, must be exerted on the ground. The magnitude of this force depends, among others, on the velocity of progression and on the radius of the circular path [28,38]. In the frame of this paper, we aim to answer four questions/hypotheses to better understand the mechanisms underlying curved running.

First, running consists of a succession of contact phases interrupted by flight phases. In the latter no lateral force acts upon the body and so the CoM movements are uniformly rectilinear in the horizontal plane [39]. When running in a curve, an "over-deflection" should occur during the contact phase to compensate for the rectilinear motion during the aerial phase and keep the runner on the circular trajectory. This over-deflection should be countered by an increased braking fore-aft force which prevents inappropriate rotation, thus allowing the runner to maintain the running trajectory [31].

A second question concerns a possible asymmetry between the inner and outer lower limbs. Indeed, while turning, the runner feels an apparent centrifugal force directed outwards, causing the subject to lean inwards to maintain balance [28]. Consequently, one might expect that the two lower limbs exert different lateral forces since the inner limb is pushing on the lateral side of the foot, whereas the outer limb pushes with the medial side of the foot. This asymmetry has been described when running at sprinting speeds on an athletic track [20]. The question arises as to which extent this asymmetry also persists at slower speeds and shorter radii of curvature?

The third goal of this paper is to study the effect of running speed and radius of curvature on the muscular work ($W_{com}$) done while running on a circular arc. To identify the sources of $W_{com}$, the work done to sustain the forward movements of the CoM ($W_Y$), the work done to sustain its vertical movements ($W_z$), and the work done to modify the direction of progression ($W_X$) were also measured.

When running on a straight line (SL), $W_{com}$ is assessed from the forces exerted by the ground under the foot in the fore aft (*y*-axis), lateral (*x*-axis) and vertical (*z*-axis) directions

vertical component of the velocity of the CoM in the $o$-$x$-$y$-$z$ reference frame; $v_X$, $v_Y$, $v_z$, lateral, fore aft and vertical component of the velocity of the CoM in the $O$-$X$-$Y$-$z$ reference frame; $v_h$, Velocity vector of the CoM in the transverse plane; $W_{com}$, Work done to sustain the movements of the CoM relative to the surroundings; Wcom+, Positive work done to sustain the movements of the CoM relative to the surroundings; WX+,WY+,Wz+, Positive work necessary to sustain movements of the CoM respectively in the $X$-, $Y$- and $z$-directions. The positive and negative superscripts indicate the increments of work; $x_0$, $y_0$, Coordinates of the centre of the circle in the he $o$-$x$-$y$-$z$ reference frame, calculated from a circular regression.

[1,40]. When running along a curve, using a fixed referential attached to the laboratory makes no physiological sense since the fore aft and lateral directions no longer correspond to the $y$- and $x$-directions. It is therefore necessary to use a referential which rotates with the runner. A first solution should be to rotate the frame of reference at each instant so that the $Y$-axis of the new reference frame is tangent to the circular trajectory that the runner follows and the $X$-axis is pointed towards the centre of the circle. However, in this case, it is difficult to separate the work done to move the CoM in the axis of progression ($W_Y$) from the work necessary to modify the direction of progression ($W_X$) (see supplementary material).

As the foot is fixed on the ground during most of the contact phase [3] (S4 Fig), we propose to use a referential that rotates only once per step: at each foot contact. The origin of the referential is then placed at the position where the foot touches the ground and is oriented in a way where one of the axes is in the direction of the velocity vector in the transverse plane at the instant of touch down. In this new $O$-$X$-$Y$-$z$ referential, the $Y$-axis lies in the direction of progression and the $X$-axis points inwards. In this referential frame, $W_Y$ corresponds to the work done to sustain the movements of the CoM in the fore-aft direction and $W_X$ to the work done to sustain the lateral movements of the CoM in the referential of the runner.

Fourth, when running on a straight line most of the movement is done in the sagittal plane and the work to sustain the lateral movements of the CoM is negligible [16,41]. The energy of the CoM due to its fore-aft velocity ($E_y$) and the energy due to its vertical movements ($E_z$) are in phase [39]. Part of the kinetic and potential energy lost during the braking phase can be stored to be reused to accelerate and elevate the CoM during the second part of contact, as in a spring-mass system bouncing on the ground [39]. When turning, the work to sustain the lateral movements of the CoM is no longer negligible since a lateral force must be exerted during the entire contact phase to deflect the trajectory of the CoM. Therefore, part of the kinetic and potential energy lost during the first part of contact when the CoM is decelerated and lowered in the sagittal plane (braking phase) could be used to redirect the CoM in the lateral direction.

To answer these four questions, the forces that the ground exerts under the feet and the movements of the pelvis were recorded during 1–4 strides (2–8 steps) while running in curves where the radius of curvature was of either 6 m or 18 m, at different speeds ranging from 2 to 6 m s$^{-1}$. Based on these data, the trajectory of the CoM and its energy were computed. From there, $W_X$, $W_Y$, $W_z$ and $W_{com}$ were assessed. Results were compared with those obtained while running on a straight line.

## Materials and methods

### Participants and experimental procedure

Eleven recreational healthy male runners (height: 1.81 ± 0.03m, mass: 74.91 ± 5.38 kg, age: 24.64 ±2.91 years, mean ± SD) participated in the study. The participants were all amateur-level runners with 10km personal bests ranging from 33–45 minutes and were recruited through informal person-to-person communication channels. Informed written consent was obtained, and the study followed the guidelines of the Declaration of Helsinki. All procedures were accepted by the UCLouvain Ethical Committee (B403201940199). Power analysis was performed using GLIMMPSE 3.1.2. [42] (Denver Colorado, USA) to estimate a sufficient number of participants required to avoid type II error (Power 1-ß: 0.8, minimum sample size: 9 participants).

Participants took part in two sessions 6 weeks apart. First, they were asked to run at different speeds, ranging from 2 to 6 m s$^{-1}$ (~7 to 22 km h$^{-1}$), on a circular path (radius of curvature 18 m) instrumented with 16 force plates. Second, they were asked to run on a 6 m radius curved path, and on a treadmill, for comparison with SL. The equipment used to record data is

explained in the section below. The six-week period interval between the two data collection sessions was necessary to reconfigure the force plates.

Participants ran in a 20 m long straight corridor before arriving in the curved path. They took at least one step in the curve before the measurements were taken. Participants were first asked to run at intermediate speeds (around 12–14 km h$^{-1}$), then at fast speeds (between 16 and 18 km h$^{-1}$), followed by slow speeds (between 8 and 10 km h$^{-1}$), and finally at the maximum speed that they could reach. After each trial, the average velocity of the runner, $v_h$ (explained below) and number valid of strides were checked and communicated to the participant. Trials were considered valid if they contained at least one stride (two successive steps) where $v_h$ remained constant, *i.e.*, if the sum of the decrements of $v_h$ differed less than 25% of the increments over a complete stride. Rest periods were allowed between the trials, at the runner's convenience, to avoid fatigue-bias.

**Experimental set-up.** During the experiments, two types of variables were measured. First, the reaction forces that the ground exerts under the foot (GRF) were evaluated using force platforms (kinetic variables). Second, the movements of the pelvis were recorded using high-speed infrared cameras (kinematic variables). Both systems are described in length below. Two pairs of photocells were placed at the level of the neck, ~6 m apart, at each end of the running track. These cells both triggered and stopped the acquisition of kinetic and kinematic data.

Measurement of kinetic variables during curved running

For the curved running sessions, 16 custom built force plates of 1 m x 1 m each were placed side by side in the configurations illustrated in Figs 1A and 2A. The circular paths bended to the right. To guide the runner during curved running, a 0.5 m wide lane was delimited by two white stripes. The characteristics of the force plates are described in detail in Genin et al. [43] and are only presented briefly here. Each plate measured the three components of the GRF by means of four force transducers placed under each corner of the plate surface. Each plate possessed its own data acquisition system comprising amplifiers, anti-aliasing low-pass filters (4-pole Bessel filter with a -3 dB cut-off frequency of 200 Hz), a 16-bit analogue-to-digital converter sampling at 1 kHz and a micro-controller (Rabbit Semiconductor, Davis, CA, USA). The 16 plates were connected to a central PC *via* ethernet using the TCP/IP protocol.

*Measurement of kinetic variables during straight-line running*. During SL, the GRF were measured by means of an instrumented treadmill, which consisted of a modified commercial treadmill (h/p/Comos-Stellar, Germany) mounted on four force transducers (Arsalis, Belgium). Since the whole body of the treadmill (including the motor) was mounted on the transducers, these were measuring the three components of the GRF exerted by the treadmill under the foot [44]. A more in-depth description of the treadmill can be found here [3]. The signals were amplified, low-pass filtered (4-pole Bessel filter with a -3 dB cut-off frequency at 200 Hz) and digitized by a 16-bit analogue-to-digital converter (National Instrument, PCI-MIO-16E-4) at 1000 Hz.

*Measurement of kinematic variables*. Bilateral, full-body three-dimensional (3D) kinematics were recorded by means of a Qualisys system (Gothenburg, Sweden) equipped with 12 Mocap OQUS 6+ cameras placed around the running track or the treadmill, plus one video Miqus M1 camera. Participants were equipped with 43 retro-reflective markers glued onto the skin according to the Qualisys sports marker set disposition. In the frame of this study, only the markers placed on the left and right anterior superior iliac tuberosity (SAT$_L$ and SAT$_R$) and at the level of the second sacral vertebra (S2) were used. Kinematics were recorded at a sampling rate of 100–240 Hz and oversampled during post-processing at 1 kHz to match the sampling frequency of the kinetic data. An oversampling of the kinematic data was chosen rather than a down sampling of the kinetic data to ensure that the force signal characteristics were not lost during data-processing.

A

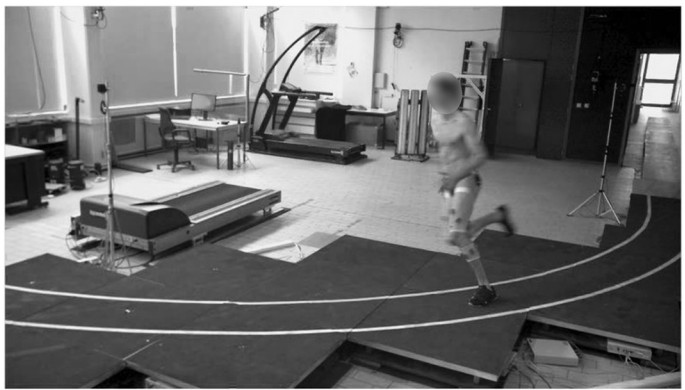

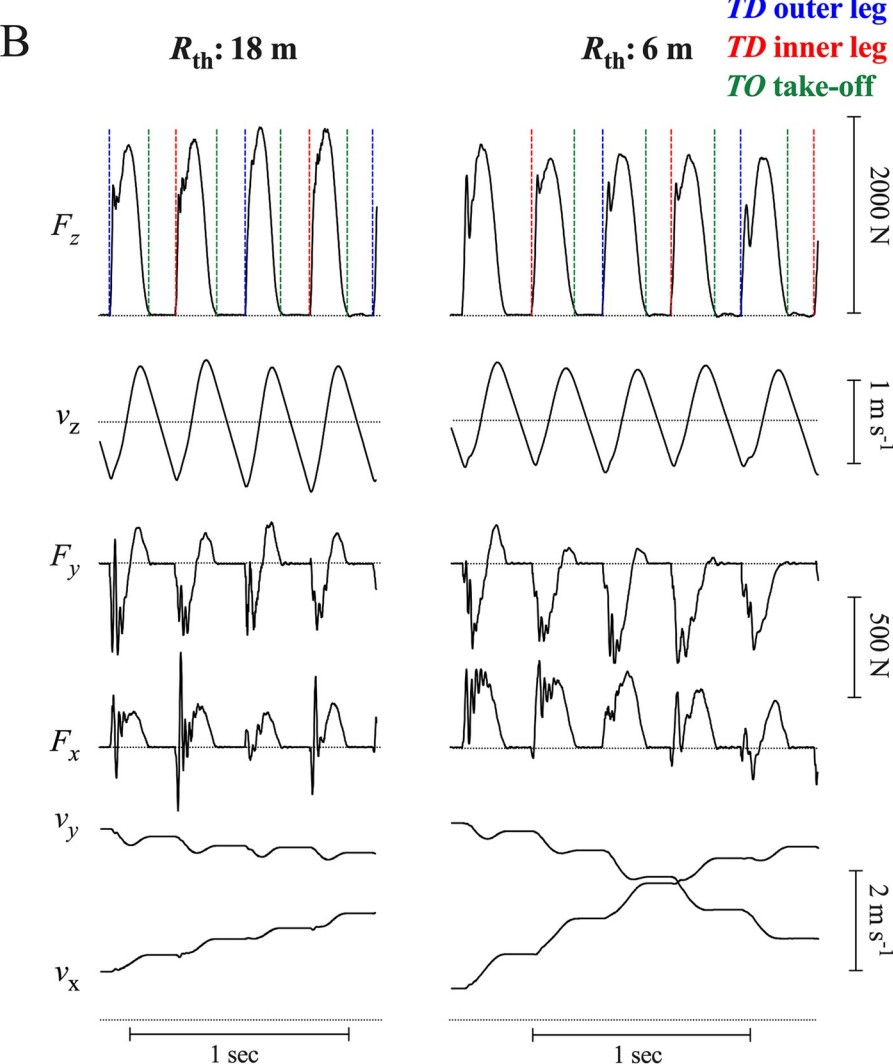

**Fig 1. Image of the set-up and typical traces. Panel A:** Image of the 16 (1x1 m) force plate set-up for the 6 m radius of curvature. A 0.5 m track is drawn over the to guide the runner. **Panel B:** Typical traces of the three components of the GRF and of the velocity of the CoM relative an inertial reference frame attached to the lab. For both radii of curvature, variables were recorded over entire trials on one subject (body mass: 65.1 kg, height: 1.78 m) running at 14 km h$^{-1}$. From top to bottom: Vertical component of the GRF ($F_z$) and of the velocity ($v_z$), components of GRF in the transverse

plane along the $y$- and the $x$-axis (respectively $F_y$ and $F_x$). Further below, the two components of the horizontal velocities along the $y$-axis ($v_y$) and the $x$-axis ($v_x$). The dashed vertical lines drawn over $F_z$ correspond to the time of touchdown of the inner leg (red) and of the outer leg (blue) and of the time of take-off of both legs (green).

**Data processing.** In this manuscript, we used two different reference frames. The first *o-x-y-z* is fixed in the laboratory and was used to compute and analyse the trajectory of the CoM. The second *O-X-Y-z* changed orientations at each foot contact and was used to assess the work done to move the CoM relative to the surroundings.

*Measurement of the contact and aerial period.* The lower part of Fig 1B illustrates the time-curves of the vertical ($F_z$), fore-aft ($F_y$) and medio-lateral ($F_x$) components of the GRF measured between the two photocells while running on an 18 m (left) and 6 m curve (right). The contact phase ($t_c$) corresponds to the period during which $F_z \geq 30 N$ and the aerial phase ($t_a$) to the remaining period. The dotted lines drawn on $F_z$ indicate the heel-strike of the outer leg (blue), the heel-strike of the inner leg (red) and the take-off instant (green).

*Assessment of the centre of mass movement in the transverse plane.* The CoM displacement in the transverse plane was computed from the position of a point ($PL_c$) placed approximatively in the centre of the pelvis. Since the movements of the trunk relative to those of the CoM are small [1], one can consider that the displacement of $PL_c$ is similar to the displacement of the CoM. (see S1 Fig).

At each instant, the position of $PL_c$ in the transverse plane (*i.e.*, the centroid of the triangle) was computed from the $SAT_R$, $SAT_L$ and S2 markers (Fig 2B, upper right corner):

$$PL_{cx} = \frac{SAT_{Rx} + SAT_{Lx} + S2_x}{3} \qquad \text{and} \qquad PL_{cy} = \frac{SAT_{Ry} + SAT_{Ly} + S2_y}{3}. \qquad (1)$$

The displacement of the CoM can also be calculated *via* the double integration of the GRF. This signal gives the displacement and not the position of the CoM. To obtain the CoM's position, one has to use integration constants, which require knowing the initial and final position of the runner over the recorded distance. When running in a straight line, the positions of the start and end triggers can be used as the integration constant [2]. However, although the width of the lane in which the subject was asked to stay in was relatively small as compared to the regulatory width of an athletic track lane (1.22 m [45]), if the runner began on the inside or outside of the lane could change the distance covered in either direction considerably.

For the same reason, while the runner's average trajectory should be equal to the theoretical radius ($R_{th}$) determined by the running lane curvature (*i.e.*, 18 m or 6 m), they could deviate from this desired radius of curvature. As a result, the radius of curvature based on the runner's trajectory was measured in each trial by fitting a circular regression to the ($PL_{cx}$, $PL_{cy}$) coordinates over the strides analysed to obtain the actual radius of curvature ($R$) and the position of the centre of the circle ($x_o$, $y_o$). An example of the difference between the $R$-curve and the $R_{th}$-curve is illustrated on the diagram of the force-plates in Fig 2A. For each trial, the difference between $R$ and the theoretical radius ($R_{th}$ = 6 or 18 m) was computed.

To assess the oscillation of the CoM around the circular trajectory, the distance between $PL_c$ and the centre of the circle was calculated at each instant of the stride. The difference between this "instantaneous" radius of curvature ($r$) and the radius $R$ was thus obtained by:

$$r - R = \sqrt{(PLC_{cx} - x_o)^2 + (PLC_{cy} - y_o)^2} - R. \qquad (2)$$

*Computation of acceleration velocity and displacement of the centre of mass.* Below we presented the method to compute the acceleration, velocity and displacement of the CoM from

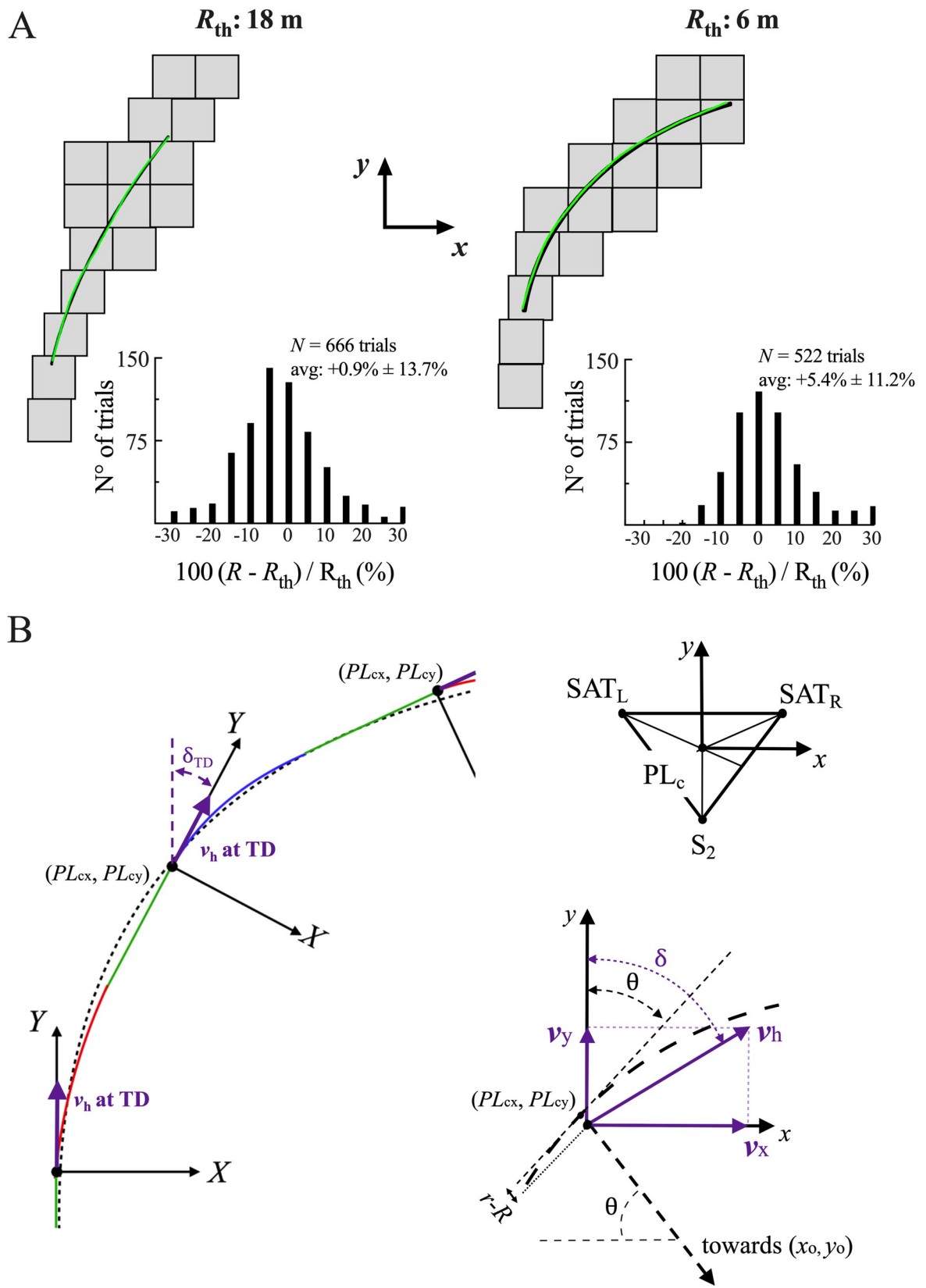

**Fig 2. Force plate disposition and experimental considerations. Panel A:** Disposition of the 16 (1x1 m) force plates for both the 18 m and 6 m radii of curvature. For both radii of curvature, the theoretical radius $R_{th}$ (black line) is drawn over the diagram of the force platforms superimposed with the trajectory computed from radius $R$ (green line). Note, $R$ is obtained by fitting a circular curve through the points $PL_c$. The frequency distributions of $(R-R_{th})/R_{th}$ is shown beside this as a histogram. **Panel B:** The upper left diagram of the panel shows the position of $PL_c$ relative to the three markers glued on the skin of the subject (*see Assessment of the movement of the CoM in the transverse plane*, in the Methods section). The lower left diagram illustrates how the two angles $\delta$ and $\theta$ are computed (see Eqs 7 and 8). The left diagram illustrates how the referential O-X-Y-z is rotated at each foot contact (see *Computation of the work*, in the Methods section).

the force platform signal relative to an inertial reference frame fixed to the lab (the method to compute these variables on a treadmill are presented in Gosseye et al. [46]).

The acceleration components in the three directions of space were obtained respectively as:

$$a_x = \frac{F_x}{m}, \qquad a_y = \frac{F_y}{m} \qquad \text{and} \qquad a_z = \frac{F_z - BW}{m}, \tag{3}$$

where $m$ is the mass of the body and $BW$ is its weight.

A time-integration of $a_x$, $a_y$, and $a_z$ between $t_1$, the beginning and $t_2$, the end of the analysed stride(s) gave the velocity changes of the CoM in the three directions of space plus integration constants (Fig 1B):

$$v_x = \int_{t_1}^{t2} a_x \mathrm{d}t + c_x,$$

$$v_y = \int_{t_1}^{t2} a_y \mathrm{d}t + c_y,$$

$$v_z = \int_{t_1}^{t2} a_z \mathrm{d}t + c_z. \tag{4}$$

Since the subject moved on the level, the average vertical velocity of the CoM over a complete number of strides is equal to zero and the integration constant $c_z$ is nil. Next, the vertical displacement ($S_z$) of the CoM was computed by time-integration of $v_z$.

The integration constants $c_x$ and $c_y$ were determined by dividing the distance travelled by the point $PL_c$ during the strides analysed in the $x$- and $y$-direction (respectively $\Delta PL_{cx}$ and $\Delta PL_{cy}$) by the duration of the strides ($T_{tot}$):

$$c_x = \Delta PL_{cx}/T_{tot} \qquad \text{and} \qquad c_y = \Delta PL_{cy}/T_{tot}. \tag{5}$$

At each instant, the magnitude of the velocity vector in the horizontal plane ($v_h$) was computed as:

$$v_h = \sqrt{v_x^2 + v_y^2}. \tag{6}$$

and its orientation relative to the inertial reference frame of the lab was measured as (Fig 2B, lower right corner):

$$\delta = \text{atan}\frac{v_x}{v_y}. \tag{7}$$

The angle $\grave{e}$ between the tangent of the circle and the *x*-axis of the lab's referential was also measured:

$$\theta = 180° - \text{atan} \, \frac{PL_{cy} - y_o}{PL_{cx} - x_o}. \tag{8}$$

If $\ddot{a} = \grave{e}$, $v_h$ had the same direction as the tangent of the circle. If $\ddot{a} < \grave{e}$, $v_h$ was directed outside the trajectory of the circular path and if $\ddot{a} > \grave{e}$, $v_h$ looked inwards.

**Computation of the work.** Here we presented a method to assess the muscular work done each step to move the CoM ($W_{com}$) while running in a curve. To validate our method, we also computed $E_{com}$ in two other ways. These results are presented in the supplementary material (see S2 Fig) and show that $E_{com}$ gives identical values for the three methods. However, the method chosen here allowed to better identify the different sources of work in $W_{com}$, *i.e.*, to distinguish between the work done to move the CoM forwards, upwards and to change its direction in the transverse plane.

At the last instant prior to the contact phase, the reference frame was rotated around the vertical *z*-axis so that the *Y*-axis of the new reference frame corresponded to the direction of the velocity vector $v_h$ at touch-down (*TD*), which was roughly the direction of $v_h$ during the previous aerial phase (Fig 2B, left). The *X*-axis was perpendicular to that direction. During the contact phase, the reference frame was maintained fixed until the next *TD*. Placing the reference frame relative to the runner allowed to separately assess the work done during the contact phase to maintain the movements of the CoM in the direction of progression ($W_Y$), the work done to deflect the trajectory of the CoM laterally ($W_X$) and the work necessary to move the CoM vertically ($W_z$).

$\delta_{TD}$ was defined as the angle at *TD* between the velocity vector $v_h$ and the reference frame of the lab (Eq 7 and Fig 2B). The components of the GRF ($F_X$, $F_Y$) and of $v_h$ ($v_X$, $v_Y$) in the transverse plane relative to the new reference frame are computed at each instant of the step by:

$$\begin{pmatrix} F_X \\ F_Y \end{pmatrix} = \begin{pmatrix} F_x \cos \delta_{TD} - F_y \sin \delta_{TD} \\ F_x \sin \delta_{TD} + F_y \cos \delta_{TD} \end{pmatrix}, \tag{9}$$

and

$$\begin{pmatrix} v_X \\ v_Y \end{pmatrix} = \begin{pmatrix} v_x \cos \delta_{TD} - v_y \sin \delta_{TD}, \\ v_x \sin \delta_{TD} + v_y \cos \delta_{TD} \end{pmatrix}. \tag{10}$$

The energy of the CoM due to the velocity changes in the *X*-, the *Y*- and the *z*-directions are computed as:

$$E_X = \int F_X \, v_X \, \mathrm{d}t = \frac{1}{2} \, m \, v_X^{\,2}, \tag{11}$$

$$E_Y = \int F_Y \, v_Y \, \mathrm{d}t = \frac{1}{2} \, m \, v_Y^{\,2}, \tag{12}$$

$$E_z = \int (F_z - m \, g) v_z \, \mathrm{d}t = \frac{1}{2} \, m \, v_z^2 + m \, g \, S_z, \tag{13}$$

and the total energy of the CoM by:

$$E_{com} = E_X + E_Y + E_z. \qquad (14)$$

The positive and negative work done, $W_{com}^+$ and $W_{com}^-$ were respectively the sum of the positive and negative increments of the $E_{com}$-curve. To minimise errors due to noise, the increments in mechanical energy were considered to represent positive/negative work done only if the time between two successive maxima/minima was greater than 20 ms. Similarly, $W_Y^+$ and $W_Y^-$, which represent roughly the work to sustain the movements of the CoM in the direction of progression were computed from the positive and negative increments of the $E_Y$-curve; and $W_X^+$ and $W_X^-$ represent roughly the work done to deflect the CoM laterally were evaluated from the positive and negative increments of the $E_X$-curve. Finally, the work $W_z^+$ and $W_z^-$ done to move the CoM vertically were computed from the positive and negative increments of the $E_z$-curve. The mass specific and step normalised positive ($W_Y^+$, $W_X^+$, $W_z^+$ and $W_{com}^+$) and negative ($W_Y^-$, $W_X^-$, $W_z^-$ and $W_{com}^-$) works were normalised by the subject mass and step length [3,35,39]. Energy transduction between the $E_X$-, $E_Y$- and $E_z$-curves were assessed as *%Recovery* [12]:

$$\%Recovery = 100\left[1 - \frac{W_{com}^+ + |W_{com}^-|}{W_l^+ + |W_l^-| + W_f^+ + |W_f^-| + W_z^+ + |W_z^-|}\right]. \qquad (15)$$

**Selection of the analysed steps.** Over the strides selected, care was taken to select inner and outer steps where the subject was not accelerating nor decelerating and where the vertical velocity was not drifting. Therefore, the magnitude of the velocity vector in the horizontal plane ($v_h$–Eq 6) and in the vertical direction ($v_z$–Eq 4) were checked before analysis. Steps were considered suitable for analysis if the sum of the increments and the sum of the decrements of $v_h$ and of $v_z$ differed less than 25% [12]. In total, 4863 steps were analysed: 2545 outer steps and 2318 inner steps from a total of 6380 steps (3152 inner and 3228 outer steps).

**Statistics.** For each participant, the different parameters were first averaged for all left and right steps over one trial. Then descriptive statistical analysis was performed. First, a Shapiro-Wilk test was performed to verify normality. A linear mixed effect model with *Bonferroni* post-hoc correction was used to assess the individual and interaction effects of radius and speed on the calculated variables with IBM SPSS Statistics (PASW Statistics, 19, SPSS, IBM, Armonk, NY, USA). To analyse the eventual inner-outer limb asymmetry between the variables a paired t-test (t) was used, along with a Cohen's d (d) value to analyse the effect size and interpreted as follows: < 0.19 = trivial, 0.20–0.59 = small, 0.60–1.19 = moderate, 1.20–1.99 = large, > 2.00 = very large [47]. When figures were presented by speed-class, the inter-subject grand means were shown at the following speed classes ([6,8), [8, 10), [10,12), [12, 14), [14, 16), [16, 18), [18, 20), [20, 22)), these groups were only used for the figures and not used for statistical tests.

## Results

### Actual radius of curvature of the runner

Although the width of the lane in which participants were asked to run in was small (0.5m) as compared to that of an athletic track, runners could deviate from the desired radius of curvature. The distribution of the actual radius of curvature ($R$) compared to $R_{th}$ was quantified and presented in Fig 2A. At 18 m, $R$ was on average 0.9% ± 13.7% higher than the $R_{th}$. At 6 m, $R$ was on average 5.4% ± 11.2% higher than $R_{th}$. Across all trials, the coefficient of determination ($r^2$) of the curved regression fit was of 0.99 and the RMSE was of 0.015 m on average.

## Deviation from a circular trajectory

Note that despite the high coefficient of determination, the trajectory of the CoM deviates from the perfect circular path (Fig 3 left) and it exhibits a roughly similar progression at all speeds on the two curvatures (S3 Fig). The right column of Fig 3 presents the evolution of the angles $\delta$ and $\theta$ during the stride, which also present a similar pattern in all situations.

At TD on the inner leg, the CoM is located outside of the curve as indicated by the positive $r$-$R$ (Figs 3 left and 4B), whereas at TD on the outer leg, $r$-$R$ is negative, indicating that the CoM is located inside the curve (t = -30.1, p <0.001, d = -0.9). When speed increases, $|r$-$R|$ at TD increases (F = 9.315, p <0.001) both on the inner and outer leg.

Fig 4C presents the differences $\delta_{TO}$-$\delta_{TD}$ and $\theta_{TO}$-$\theta_{TD}$. Except at very low speeds ($\leq$ 8 km h$^{-1}$), $\delta_{TO}$-$\delta_{TD}$ > $\theta_{TO}$-$\theta_{TD}$. The difference $\delta_{TO}$-$\delta_{TD}$ increases with speed (F = 79.9, p<0.001) and when the curvature decreases (F = 1455.6, p <0.001). Furthermore, $\delta_{TO}$-$\delta_{TD}$ is greater for the inner-leg as compared to the outer leg (t = -10.6, p <0.001). Note that when the body touches the ground, the vector $v_h$ is increasingly directed outside the circle ($\delta$ - $\theta$ < 0) at all speeds (on average at 8 km h$^{-1}$, $\delta_{TD}$ - $\theta_{TD}$ ~ -1° and ~ -3° at 22 km h$^{-1}$; speed: F = 21.52, p<0.001), except at TD on the inner leg when speeds $\leq$ 8 km h$^{-1}$. When the body leaves the ground, $v_h$ is directed inside the circle at all speeds and curvatures (speed: F = 44.6, p<0.001; radius: F = 43.8, p<0.001), again except for the slowest speeds.

**Spatiotemporal parameters.** Fig 4A shows the effect of running speed and curvature on the contact ($t_c$) aerial ($t_a$) and step ($T$) period. Regarding the effect of speed, below 10 km h$^{-1}$, $t_a$ represents less than 30% of $T$. When speed increases, $t_c$ decreases (F = 166.6, p <0.001) whereas $t_a$ increases (F = 60.9, p <0.001). At speeds above 15 km h$^{-1}$, both $t_a$ and $t_c$ decrease. On an 18 m curvature, $t_a$ is approximately equal to $t_c$, whereas on a 6 m curvature, $t_a$ is less than $t_c$.

Regarding the effect of the curvature, when running on a circular path, $t_c$ is slightly reduced as compared to $t_c$ during SL (F = 4.86, p = 0.01), however no difference between the two radii is evidenced (*post-hoc* p = 0.335). On the contrary, a significant effect of the curvature on $t_a$ is observed (F = 8.3, p = 0.002): $t_a$ is smaller at 6 m as compared to 18 m (*post-hoc* p = 0.008). This last difference is mainly highlighted at high speeds.

**Work to move the centre of mass relative to the surroundings.** Fig 5A presents the three components of the GRF signal relative to the *O-X-Y-z* reference frame of steps of a subject running at 14 km h$^{-1}$ (same subject than in Fig 1). The *X*-component (which is always directed towards the inside of the curve) increases when the radius of curvature decreases. On a 6m radius of curvature, a difference between the inner and the outer limb occurs.

Fig 5B presents the energy changes due to the movements in the three directions of space plus the total energy of the CoM during the same steps as in Fig 5A. The increments of the $E_{com}$-curves represent the positive work done to move the CoM ($W_{com}^+$), whereas the increments of $E_z$, $E_X$ and $E_Y$ represent the positive work necessary to move the CoM along the *z*-, *X*- and *Y*-axes (respectively $W_z^+$, $W_X^+$ and $W_Y^+$).

The two panels on last row of Fig 6 show the effect of speed and of radius of curvature on $W_{com}^+$: both variables have a significant effect on $W_{com}^+$ (speed: F = 7.05, p <0.001 radius: F = 23.2, p = 0.002). $W_{com}^+$ decreases with speed in all curves. Whereas, when comparing running on a curve to SL, $W_{com}^+$ is greater on a 6 m radius curve than during SL (*post-hoc* p < 0.001) and is not significantly different on an 18 m radius curve compared to SL (Fig 6, lower panels).

The work $W_z^+$ decreases when running speed increases (F = 320.3, p<0.001) but is not modified by the radius of curvature (F = 2, p = 0.156) (Fig 6 second row from above). $W_X^+$ is negligible during SL (S2 Fig), whereas, when running on a curve, $W_X^+$ increases both with

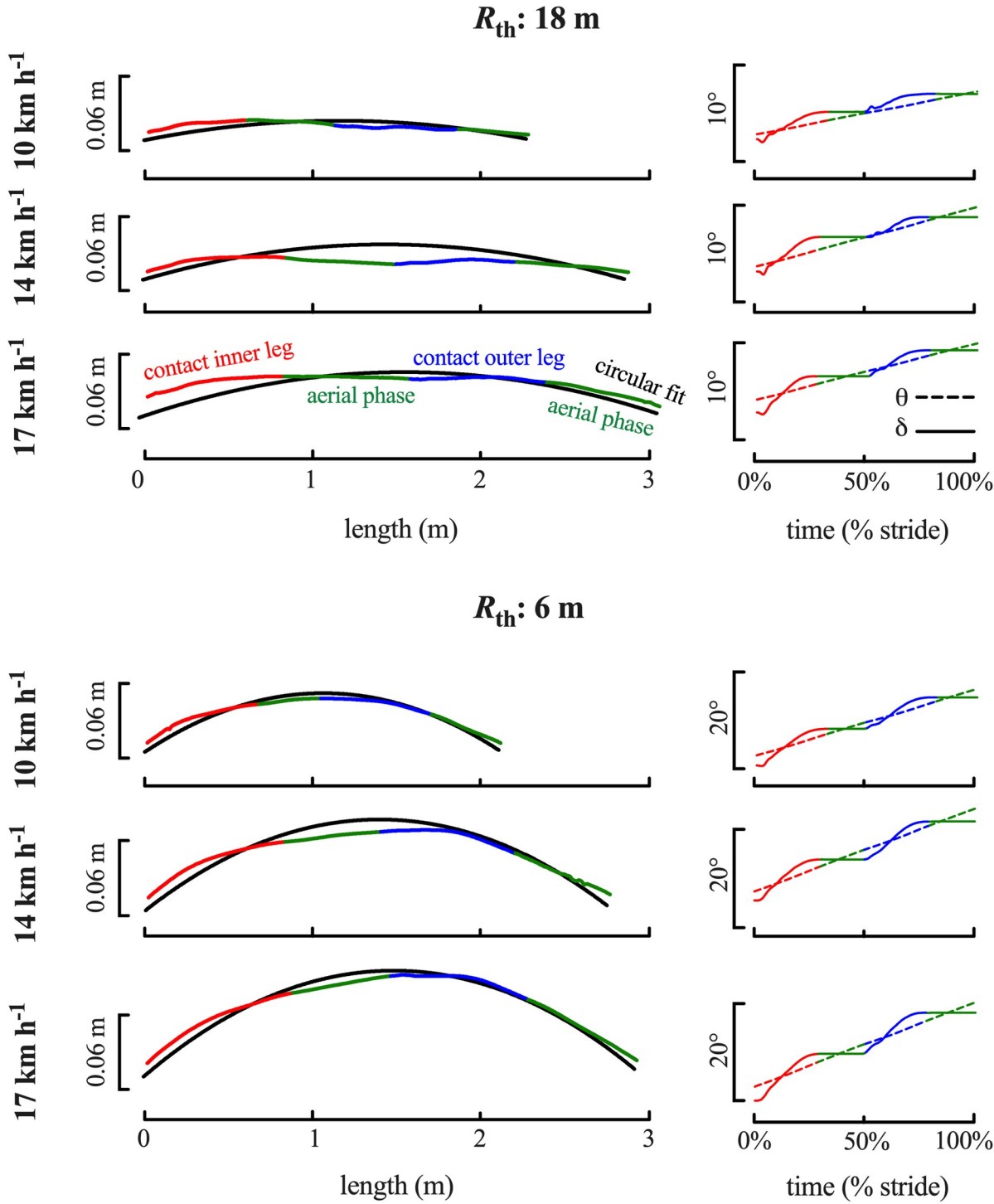

**Fig 3. $PL_c$ oscillation around the circular fit and angle deviations.** The left panels illustrate the typical traces of the oscillation of $PL_c$ over one stride around the circular fit $R$ (black line), for the same subject as in Fig 1. In each panel, the circular path is rotated so that the cord of the arch defined by the first and last point of the stride is horizontal. The vertical scale is amplified to accentuate the oscillations around the curve. The right panels represent the temporal changes of the angles $\delta$ (continuous line) and $\theta$ (interrupted line) over the strides presented in the left panel. Colours are the same as in Fig 1.

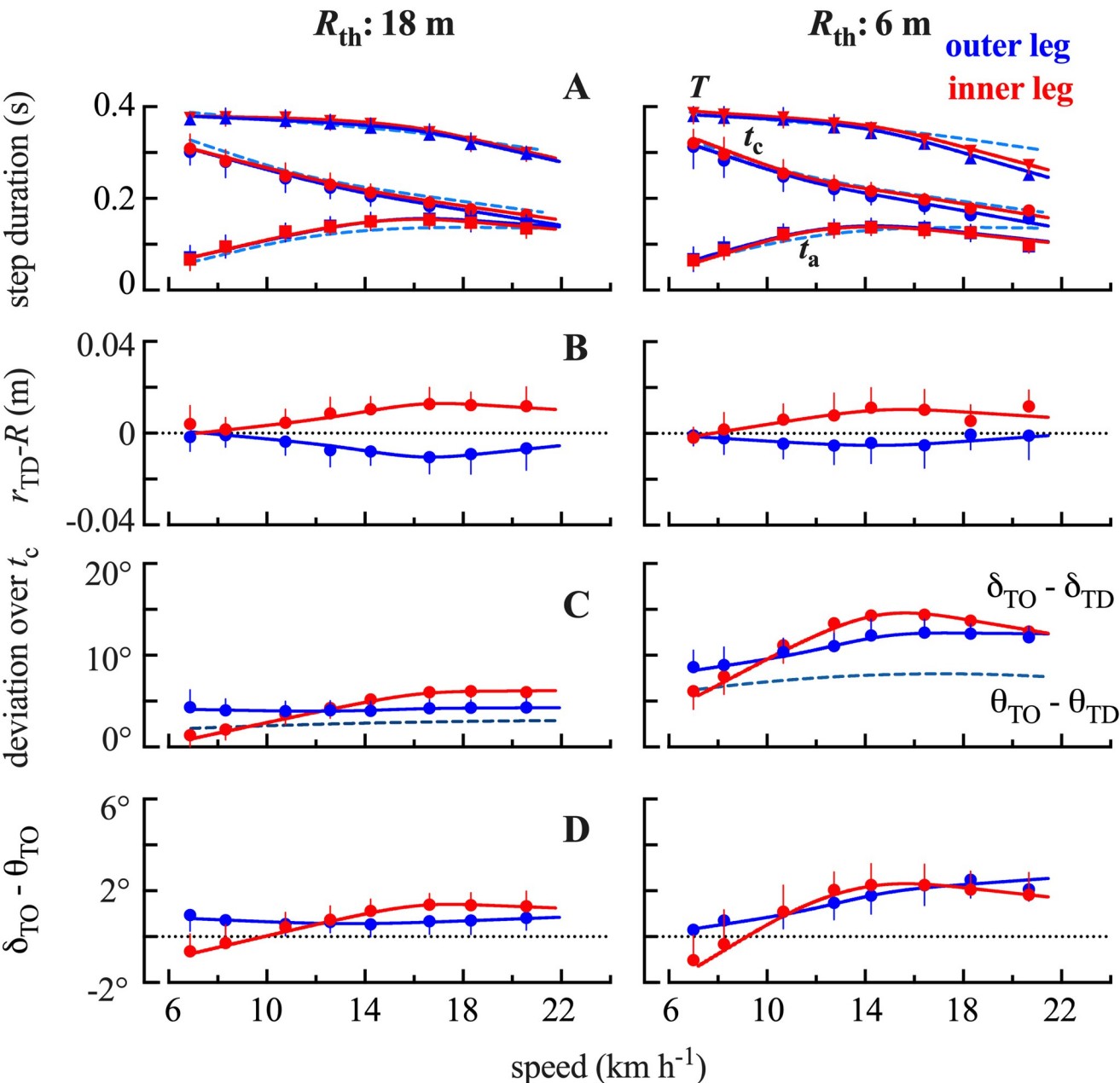

**Fig 4. Comparisons between running at 18 m, 6 m and on a straight line. Panel A**: Effect of running speed on the time of contact ($t_c$) and aerial time ($t_a$) of steps on of the inner (red) and outer (blue) while running a curve of 18 (left) and 6 m (right) radius. **Panel B**: Effect of speed on the difference between the "instantaneous" radius at touch down, $r_{TD}$, and the radius described by the circular fit of the $PL_c$ trajectory, $R$. **Panel C:** The difference between the angles $\delta$ (continuous line) and $\theta$ (interrupted line) at take-off (subscript TD) and touch down (subscript TO). **Panel D:** The difference between $\delta$ and $\theta$ at take-off. In each panel, the continuous lines represent spline functions fitted on all the data.

speed of progression (F = 95.2, p <0.001) and with radius of curvature (F = 2109.9, p<0.05). The work $W_Y^+$ increases with running speed (F = 40.9, p<0.001) and with radius of curvature (F = 119.1, p<0.001). This last difference is seen when comparing each radius class with another (*post-hoc* p<0.001).

Both $W_Y^+$ and $W_X^+$ are different on the inner and the outer limb ($W_Y^+$: t = 10.61, p<0.001, $W_X^+$: t = -12.3, p <0.001), with moderate effect sizes in both cases ($W_Y^+$: d = 0.53, $W_X^+$: d = -

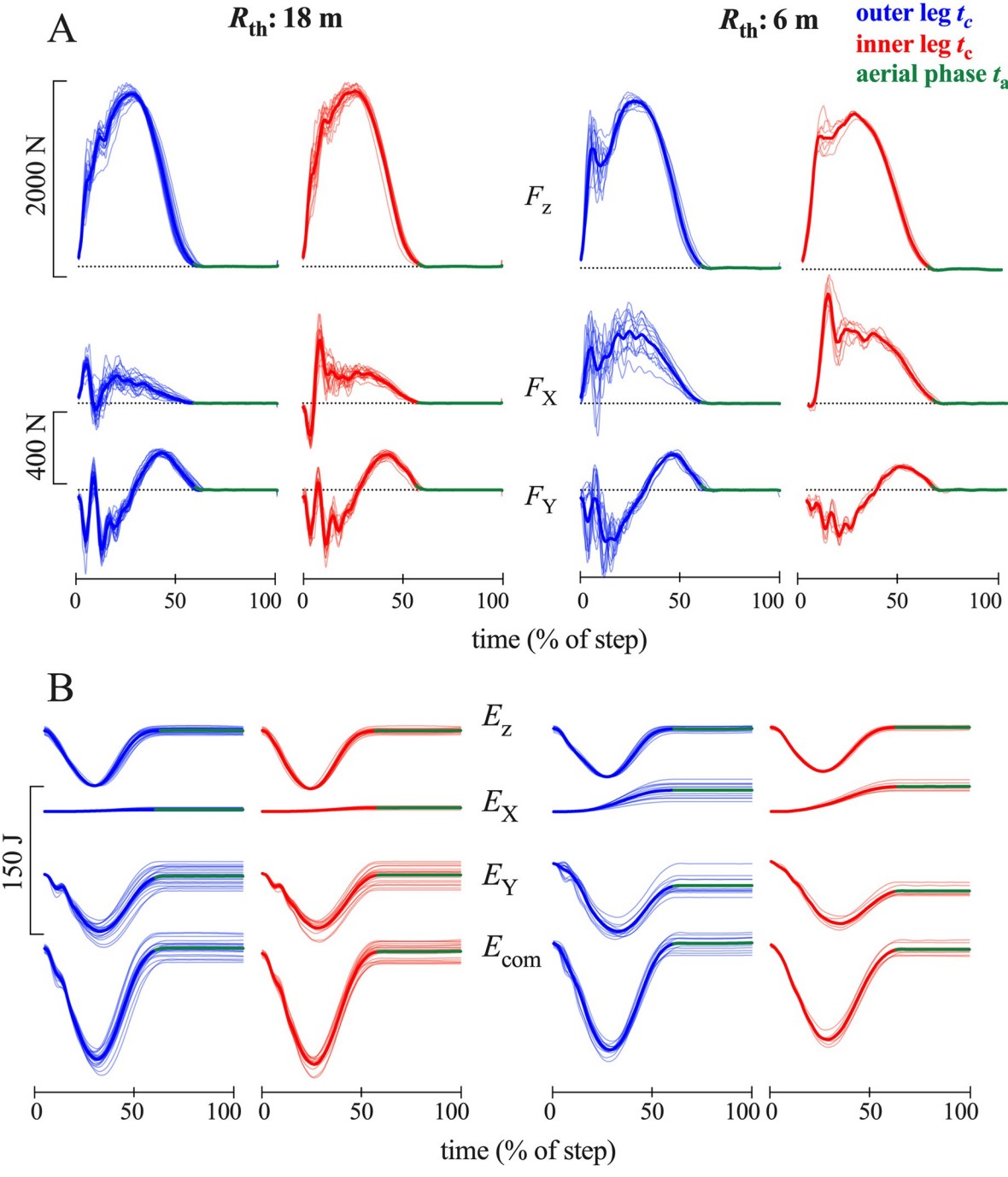

**Fig 5. Rotated referential forces and energies. Panel A:** The time-evolution of the three components of the GRF ($F_z$, $F_Y$ and $F_X$) recorded in the $O$-$X$-$Y$-$z$ inertial reference frame during one step on the outer (blue) and inner leg (red) while running at 14 km $^{-1}$ on a curve with an 18 m (left) and 6 m radii (right). The green line represents the average aerial phase for each trace. The thin lines are the individual traces recorded on the same subject than in Fig 1. The thick line is the average of all the steps. **Panel B:** Energy-time curves of the CoM on all the same steps than in panel A. $E_z$ is the potential plus kinetic energy of the CoM due to its vertical movements, $E_X$ and $E_Y$ are the kinetic energies due to the velocity of the CoM along the X- and the Y-axis, respectively. $E_{com}$ is the total energy of the CoM ($E_{com} = E_z + E_X + E_Y$). In both panels, time is expressed as a percentage of the step period.

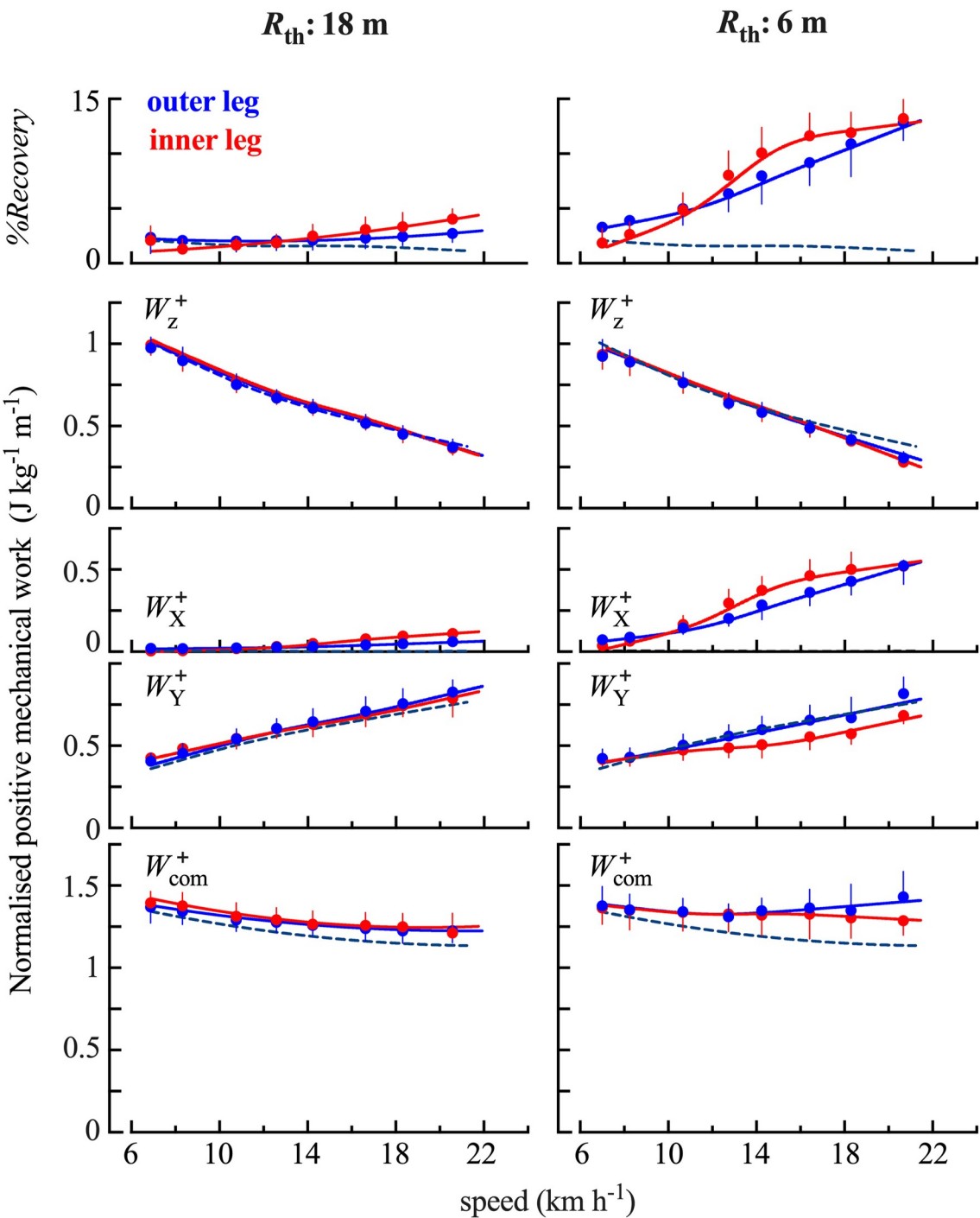

**Fig 6. %Recovery and work done to sustain the centre of mass movements.** *%Recovery* and work done on the inner (red) and outer (blue) limb for both radii of curvature as a function of speed. The upper panels present energy recovered through the transduction between $E_X$, $E_Y$ and $E_z$. The four panels below present the mass-specific positive work normalised per unit distance. From top to bottom row 2 presents the work necessary to sustain the movements of the CoM respectively in the vertical direction ($W_z$), row 3: Along the X-axis ($W_X$) and row 4: Along the Y-axis ($W_Y$). The bottom panel represents the work done to sustain the movements of the CoM relative to the surroundings. The interrupted lines represent spline fits for the left foot during SL. Other indications are as in Fig 4.

0.56). Only a small difference was observed regarding the work to move the CoM vertically ($W_z^+$: t = -6.47, p <0.001, d = -0.2), and no difference between the inner and outer limb was evidenced for $W_{com}^+$ (t = -0.47, p = 0.637).

**Transfer in energy used to deflect the centre of mass.**   The %*Recovery* (Fig 6, upper panel) increases both as a function of speed (F = 26.63, p<0.001) and as a function of radius of curvature (F = 731.8, p <0.001). On an 18 m curvature, the %*Recovery* is always lower than 5%, as in SL, whereas on a 6 m curvature at higher speeds, one can recover about 12% of energy.

## Discussion

The results of the present study provide insight into the biomechanics of running along a curve and shed light on the adaptations that occur in response to changes in speed and radius of curvature. The implications of these findings within the broader context of sport sciences will be considered upon.

When instructed to run at different speeds inside a circular path marked on the ground (Figs 1A and 2A), participants kept a circular trajectory, as evidenced by the strong coefficient of determination of the circular fit applied to the $PL_c$ trajectory ($r^2$ = 0.99). However, the mean value of the actual radius curvature ($R$) is 1% greater than the theoretical $R_{th}$ on an 18 m curvature and more than 5% greater on a 6 m curvature (histogram in Fig 2A). This means that despite the ground markings delimiting a smaller corridor than on an athletic track (0.5 m vs. 1.22 m), most participants follow a curve greater than $R_{th}$ by entering the curve on the outer limit, leaning inside to the apex and finishing close of the outer limit. This strategy reduces the detrimental effects of curvature especially at high running speeds on tight curves [28,38]. This fact is well known by trainers when they compare running performances between inside and outside lanes [36], or between outdoor and indoor venues.

Running is defined by unipedal stance phases followed by flight phases [39]. Following Newton's second law, during the flight phase the CoM describes a uniform rectilinear motion which deviates from a circular path (Fig 3). Therefore, the rectilinear motion during the aerial phases needs to be "over-compensated" during the contact phases. This is done by applying a lateral GRF which increases when the curvature decreases (Fig 5A).

In both legs, $\delta < \theta$ at TD (Fig 3 right), indicating that $v_h$ is directed outwards. Around mid-contact, $\delta$ becomes greater than $\theta$, indicating that $v_h$ is directed inwards during the second part of $t_c$. During contact, the $v_h$ vector is thus reoriented. The difference $\theta_{TO}$-$\theta_{TD}$ represents the angle covered along the circular path *i.e.*, the rotation of the tangent to the circle during $t_c$. The actual rotation of $v_h$ during $t_c$ (*i.e.*, $\delta_{TO}$-$\delta_{TD}$) is greater than $\theta_{TO}$-$\theta_{TD}$ (Fig 4C), which clearly shows that there is an "over-compensation" of the rotation during $t_c$.

At slow running speeds, as in SL [16], $t_c$ represents more than 70% of the step period (Fig 4A). As a result, the deviations due to the linear trajectory during $t_a$ are small ($\delta_{TO}$- $\delta_{TD} \simeq \theta_{TO}$-$\theta_{TD}$ Fig 4C). With increasing speed, the "over-compensation" during $t_c$ increases. It is interesting to note that at high speeds, on a 6 m curvature, $t_a$ is shortened (Fig 4A), most likely to best follow the circular path and to contain the increase of lateral force during $t_c$. A decrease in $t_a$ has been reported at very high running speeds (9.3 m s$^{-1}$) as the inner leg has "less room" (*sic*) to perform its swing phase [20]. Furthermore at high speed bend sprinting, differences in the contact period ($t_c$) between the inner and outer limb have also been reported [20]. In the present study, we did not find any statistical difference in $t_c$ between the two limbs. This apparent discrepancy is most likely due to the fact our participants did not run faster than 5.8 m s$^{-1}$ (Fig 4A).

When the body leaves the ground, the vector $v_h$ is directed towards the inside of the circle (except at TD on the inner leg with speeds $\leq$ 8 km h$^{-1}$). This strategy reduces the deviation (*r-*

$R$) from the circular path; the difference $\delta - \theta$ is positive at take-off (Fig 4D), suggesting that the runner directs the velocity vector towards the inside of the circle at the end of the support phase.

An asymmetry in the GRF between the inner and outer limb has already been described although the results sometimes seem contradictory. When running on a 1-6m curve, Chang and Kram [27] found that the outer limb produces greater peak lateral forces than the inner limb. In smaller radii the outer leg performs a cutting manoeuvre, performing greater lateral impulses than the inner leg [19,20,48]. When running at 6 m s$^{-1}$ on a radius of 31.5 m curvature, Hamill et al. [33] observed larger peak lateral forces on the inner rather than the outer leg. Churchill et al [20] found a greater inward force in the inner rather than outer limb when running at high speeds on an athletic track. In addition, athletes have shown greater muscular demand in the medial gastrocnemius of the inner than the outer leg [49].

In the present study, an asymmetry between the inner and outer leg is observed at all speeds on the two curvatures (Figs 3 left and S3). During stance on the inner limb, the CoM tends to be located outside the circular path, whereas on the outer limb, the CoM tends to be located inside. In both legs, the CoM tends to move closer to the circular trajectory during $t_c$ (S3 Fig). During the aerial phase following the inner limb contact, the CoM tends to be located inside the circular path, whereas during the aerial phase following the outer limb contact, the CoM tends to be located outside. This may be explained by the role each limb plays in the redirection of the trajectory of the runner. At very slow speeds, the rotation of the velocity vector ($\delta_{TO}-\delta_{TD}$) during $t_c$, is mostly done by the outer leg (Fig 4C). However, as speed increases, the reorientation done by the inner leg during $t_c$ becomes greater than that of the outer leg.

Where the study of the GRFs during running while turning is not novel, observing the outcomes on the CoM mechanics over multiple strides and over different radii has, to the best of our knowledge, never been explored before. Previous studies collected the GRF on a small number of force platforms (1 or 2) placed tangentially to the circle. In the present situation, where multiple force platforms were placed side-by-side over a circular trajectory, the resolution of the GRF into components that have a fixed referential has no physiological sense. Indeed, as runners lose energy in one axis which are compensated by gaining energy in another (Figs 1B and S2), neither axis provides physiological information as to what the runner is experiencing during their trial. Instead, a rotating referential that follows the direction of progression is best suited to distinguish between the forces necessary to push the runner inwards or outwards in the direction of the centre of the circle and accelerate and decelerate the runner in an axis facing the tangent of the circle at TD.

As compared to SL, $W_{com}^+$ is greater during curved running and increases as the radius decreases (Fig 6). This increase is mainly due to an augmentation of $W_X^+$, which roughly represents the work done to deflect the trajectory of the CoM laterally, while $W_z^+$ and $W_Y^+$ remain approximately the same. On a 6 m curve, at speeds above 12 km h$^{-1}$, $W_X^+$ is greater on the inner leg than on the outer leg. However, the increase in $W_X^+$ is roughly compensated by a decrease in $W_Y^+$. Consequently, the work necessary to move the CoM in the transverse plane ($\simeq W_X^+ + W_Y^+$) is approximately the same in the inner and outer limb. This observation suggests that the inner and outer limb play different roles in maintaining balance and controlling the trajectory while running in a curve: the inner limb is doing more work than the outer limb to deviate the CoM laterally whereas the outer limb is doing more work to accelerate the CoM forwards. This difference between the two limbs disappears on larger curves (18 m), at least at speeds below 22 km h$^{-1}$.

The energy recovered through the transduction between $E_X$, $E_Y$ and $E_z$ represents less than 5% of the work done in SL, similar to previous studies [12,39,46]. In running while turning,

during the first part of contact, the $E_X$-curve is no longer in phase with the $E_Y$- and $E_z$-curves (Fig 5), which allows for energy transduction. Indeed, $E_X$ increases continuously during contact, whereas the $E_Y$- and $E_z$-curves are decreasing during the first part of contact (braking phase) before increasing during the second part. Part of the energy lost during the braking phase could be recovered to increase the velocity (and thus the kinetic energy) of the CoM in the lateral direction (Fig 6). A similar phenomenon has been observed during uphill running [16]: where part of the energy lost during the braking phase is used to increase the potential energy of the CoM each step.

## Limitations

One of the limitations of this study is that SL was performed on a treadmill whereas curved running was performed on the ground. However, it was deemed the best option considering the lack of time and the amount work to change the configuration of the platform. Although this difference in data collection may slightly bias the results, the results that we obtained on SL are in agreement with previous results obtained on overground running [1,39,40].

The movements of the CoM are assimilated to the movements of the pelvis delimited by three markers glued on the skin. Though the movements of the CoM relative to the pelvis are small [1] (S1 Fig), the present study does not take these movements into account. Furthermore, it has been shown that the external work measured based on the movements of the pelvis was overestimated as compared to the measurement based on GRF [50]. Although, this is not applicable here, it is worth noting that these movements are not exactly the same.

One of the main objectives of this study is to analyse the trajectory of the CoM/pelvis, focusing particularly on the movements of this point in the transverse plane, without considering the movements of the body segments. In a near future, it would be interesting to analyse also the movements of the body segments in 3-D to understand how a change in direction of the CoM and the rotation of the body are generated by the movement of these segments.

## Conclusion and perspectives

As compared to SL, $W_{com}$ is increased by up to 25% depending on the running speed and radius of curvature. Furthermore, the in-depth analysis of the CoM trajectory highlights an "over-deflection" during the contact phase to counter the rectilinear motion of CoM during the aerial phases. We also show that the inner and outer leg play a different role in this "over-deflection": the inner leg tends to deflect the trajectory of the CoM more that the outer leg, whereas the outer leg plays a more important role in accelerating the CoM forwards.

Indeed, the identification of asymmetries between the inner and outer limbs provides insight into potential strategies for optimising performance and minimising injury risk. Future investigations should consider the limb movements, their intersegmental coordination and internal work [1] used to re-accelerate and reorient the velocity vector. These have been previously shown in cutting manoeuvres to contribute significantly in the amount of total mechanical energy done by the runner [24]. Such quantifications of these biomechanical constraints in running while turning may have practical implications in various fields, including sports performance, injury prevention, and rehabilitation.

## Supporting information

**S1 Fig. CoM and PL$_c$ displacement.**
(DOCX)

**S2 Fig. Alternative method for analysing $W_{com}$.**
(DOCX)

**S3 Fig. Average difference between 'instantaneous' radius and actual radius.**
(DOCX)

**S4 Fig. Foot angle variation during the contact phase.**
(DOCX)

## Author Contributions

**Conceptualization:** Raphael M. Mesquita, Patrick A. Willems, Giovanna Catavitello.

**Data curation:** Raphael M. Mesquita, Giovanna Catavitello.

**Formal analysis:** Raphael M. Mesquita, Patrick A. Willems, Arthur H. Dewolf, Giovanna Catavitello.

**Funding acquisition:** Patrick A. Willems, Giovanna Catavitello.

**Investigation:** Raphael M. Mesquita, Patrick A. Willems, Giovanna Catavitello.

**Methodology:** Raphael M. Mesquita, Patrick A. Willems, Giovanna Catavitello.

**Project administration:** Patrick A. Willems.

**Resources:** Raphael M. Mesquita.

**Software:** Raphael M. Mesquita, Giovanna Catavitello.

**Supervision:** Patrick A. Willems, Arthur H. Dewolf.

**Validation:** Patrick A. Willems, Arthur H. Dewolf.

**Visualization:** Raphael M. Mesquita, Arthur H. Dewolf.

**Writing – original draft:** Raphael M. Mesquita, Patrick A. Willems, Arthur H. Dewolf, Giovanna Catavitello.

**Writing – review & editing:** Raphael M. Mesquita, Patrick A. Willems, Arthur H. Dewolf, Giovanna Catavitello.

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
