## [Decision Letter · Decision Letter 0]

9 Jan 2024

PONE-D-23-40768Kinetics and mechanical work done to move the body centre of mass along a curvePLOS ONE

Dear Dr. Mesquita,

Thank you for submitting your manuscript to PLOS ONE. After careful consideration, we feel that it has merit but does not fully meet PLOS ONE’s publication criteria as it currently stands. Therefore, we invite you to submit a revised version of the manuscript that addresses the points raised during the review process.

Both reviewers were generally positive though they indicated some points that need to be clarified. In particular, the first reviewer pointed to some methodological issues and other potentially important parameters related to the mechanics of running along a curve, while the second reviewer asked to provide more information about foot placement characteristics (see their comments).

We look forward to receiving your revised manuscript.

Kind regards,

Yury Ivanenko

Academic Editor

PLOS ONE

Journal Requirements:

“This study was funded by the Fonds de la Recherche Scientifique (F.R.S-FNRS) of Belgium.”

“This study was funded by the National Fund for Scientific Research (F.N.R.S - CDR 40013847).”

Reviewers' comments:

Reviewer's Responses to Questions

**Comments to the Author**

1. Is the manuscript technically sound, and do the data support the conclusions?

Reviewer #1: Yes

Reviewer #2: Yes

2. Has the statistical analysis been performed appropriately and rigorously? 

Reviewer #1: Yes

Reviewer #2: Yes

3. Have the authors made all data underlying the findings in their manuscript fully available?

Reviewer #1: No

Reviewer #2: Yes

4. Is the manuscript presented in an intelligible fashion and written in standard English?

Reviewer #1: Yes

Reviewer #2: Yes

5. Review Comments to the Author

Reviewer #1: Main comments

# Differently of uncurving/straight locomotion, the mechanical determinants of locomotion economy and performance in curve conditions are poorly explored. The authors aimed to compare the trajectory of the CoM and the work done to maintain its movements relative to the surroundings at different running speeds and radius of curvatures.

# The external work and CoM trajectory were determined using force-platforms measuring the ground reaction forces and infrared cameras recording the movements of the pelvis.

During curves, runners increase the rotation of their trajectory during contact phases. These deviations increased when the radius of curvature decreases and speed increases. The paper is very interesting because brings out very new knowledge on a very typical condition of locomotion in daily activities and sports. Even in running modalities of athletics the curving paths are so common.

# The paper is well written and well organized. Another strength of the paper is the quality of figures, very informative.

# The reasoning for defining the inertial reference points for determining anteroposterior and mediolateral forces and mechanical work is correct.

# Although there is a somewhat ethereal discussion about the nomenclature in the literature, I strongly suggest that the authors use the classic denomination of external work (Wext) instead of mechanical work of the CoM (Wcom) used throughout the article according to a recently published review (PMID: 34197674).

# I suggest using 2 articles that can help at different points in the article:

A classic article on maneuverability and stability by Alexander (PMID: 21708705) that can be inserted in the introduction of the article.

An article showing that energy saving is specific to the type of running (straight/steady state vs curving/irregular), i.e. runners used to straight terrain are more economical in straight conditions while soccer players are more economical in irregular conditions (shuttles). This study sheds light on possible mechanisms related to this terrain-specific adaptation (PMID: 37234418).

# A weak point of the study is the fact that it does not consider internal work, which makes the analysis of running mechanics incomplete. This criticism does not prejudge the acceptance of the article, and the article as it stands already contributes a good deal of knowledge on the subject.

# I suggest that future studies check the parameters of the mass-spring system and the landing-takeoff asymmetries of the running. Some indirect evidence (such as Recovery and ta/tc relation) indicates that the elastic mechanism is disturbed in these curve conditions (not compulsory including in the paper, just a rationale).

# Please move the Participants section to the beginning of the methods. Similarly, the selection of the analyzed steps section can be between data collection and analysis.

Minor points

Ln 63 – is not

Ln 276-277 - I've probably missed this information, but please clarify how Wcom+ and Wcom- arrive at the final Wcom value (using only positive increments and normalizing by mass and stride or step length). I suggest using a current reference together to primary reference (e.g. Cavagna et al. 1963 or Cavagna et al. 1976) for this calculation.

Ln 332 - ])

Ln 372 – consider using exact p’s (P = 0.008 instead of P < 0.008) where possible.

Ln 378 – occurs instead of seems to appear.

Ln 380 – as instead of than

Ln 438 – It is

Ln 445 – did not

Reviewer #2: General comments

This manuscript aims at investigating how the trajectory of the centre of mass of the body and the work done to maintain its movements relative to the surroundings are modified as a function of running speed and radius of curvature. The aim is commendable. Authors some curve running adaptations which provide valuable insights for sports sciences, training and performance in sports with multidirectional movements. Overall, the authors manage to fulfil their aim properly.

Specific comments

In which exact time of the foot contact does the referential change and why? How do you cope with some foot rotation during stance?

(line 181-2) “The contact phase (tc) corresponds to the period during which Fz > 10% of body weight (BW)…” Why? Isn’t it too much to detect real flight time (10% of 74.91 kgf is 7.491 kgf)?

Minor comments

(line 66) … (Wcom) have not…

(l153) please, do not use acronyms in headings;

(l289) … mass: 74.91…

(l441) sic?

(l447) … circle (except…

6. PLOS authors have the option to publish the peer review history of their article (what does this mean?). If published, this will include your full peer review and any attached files.

Reviewer #1: No

Reviewer #2: No

---

## [Author Response · Author response to Decision Letter 0]

19 Jan 2024

To the editor

Thank you for submitting your manuscript to PLOS ONE. After careful consideration, we feel that it has merit but does not fully meet PLOS ONE’s publication criteria as it currently stands. Therefore, we invite you to submit a revised version of the manuscript that addresses the points raised during the review process.

Both reviewers were generally positive though they indicated some points that need to be clarified. In particular, the first reviewer pointed to some methodological issues and other potentially important parameters related to the mechanics of running along a curve, while the second reviewer asked to provide more information about foot placement characteristics (see their comments).

We would like to thank the editor for their comments. We have made the following modifications to our text in red. Please find a detailed list below and the text in its ‘red-lined’ and new form attached.

“This study was funded by the Fonds de la Recherche Scientifique (F.R.S-FNRS) of Belgium.”

“This study was funded by the National Fund for Scientific Research (F.N.R.S - CDR 40013847).”

We have removed the funding statement from the Acknowledgements Section and from the title page and have placed the correct funding information in the cover letter.

Reviewers' comments:

5. Review Comments to the Author

Reviewer #1: Main comments

# Differently of uncurving/straight locomotion, the mechanical determinants of locomotion economy and performance in curve conditions are poorly explored. The authors aimed to compare the trajectory of the CoM and the work done to maintain its movements relative to the surroundings at different running speeds and radius of curvatures.

# The external work and CoM trajectory were determined using force-platforms measuring the ground reaction forces and infrared cameras recording the movements of the pelvis. During curves, runners increase the rotation of their trajectory during contact phases. These deviations increased when the radius of curvature decreases and speed increases. The paper is very interesting because brings out very new knowledge on a very typical condition of locomotion in daily activities and sports. Even in running modalities of athletics the curving paths are so common. 

The paper is well written and well organized. Another strength of the paper is the quality of figures, very informative. The reasoning for defining the inertial reference points for determining anteroposterior and mediolateral forces and mechanical work is correct.

We would like to thank the reviewer for their comments. We hope that the following modifications are to their liking.

# Although there is a somewhat ethereal discussion about the nomenclature in the literature, I strongly suggest that the authors use the classic denomination of external work (Wext) instead of mechanical work of the CoM (Wcom) used throughout the article according to a recently published review (PMID: 34197674).

Muscular-tendon work performed during locomotion can be divided into two parts: the external work (Wext), which is the work necessary to move the centre of mass relative to its surroundings (Wcom) plus the work done on the environment (Wenv), and the internal work (Wint) (Cavagna et al., 1975, Lejeune et al., 1999). In the present situation, Wenv is essentially zero because wind resistance is negligible and the foot is not skidding on the ground.

Recently, we have reused this distinction in nomenclature when walking and running against external traction forces (Dewolf et al., 2020; Mesquita et al., 2020).

We find that this nomenclature speaks more clearly to the work that is being done and prefer keeping the Wext = Wcom + Wenv division. We have added a sentence to clarify this:

To the best of our knowledge, the trajectory and potential adaptions of the centre of mass of the body (CoM) when running along a curve and the work done by the muscle-tendon unit (MTU) to sustain its movements (Wcom) have not yet been analysed. Indeed, since wind resistance is negligible and the foot is not skidding on the ground [14,34], essentially all the work done by the MTU relative to the CoM (classically called Wext = Wcom + Wenv [34,35]) is done to sustain its movements (Wcom) rather than to move it against the environment (Wenv), therefore Wcom = Wext.

# I suggest using 2 articles that can help at different points in the article:

A classic article on manoeuvrability and stability by Alexander (PMID: 21708705) that can be inserted in the introduction of the article. An article showing that energy saving is specific to the type of running (straight/steady state vs curving/irregular), i.e., runners used to straight terrain are more economical in straight conditions while soccer players are more economical in irregular conditions (shuttles). This study sheds light on possible mechanisms related to this terrain-specific adaptation (PMID: 37234418).

Thank you. We have added both sources to our bibliography. The former, in the first sentence when mentioning that human locomotion is usually studied in steady-state conditions. The latter, was added when explaining that curving manoeuvres are studied in the context of team sports.

# A weak point of the study is the fact that it does not consider internal work, which makes the analysis of running mechanics incomplete. This criticism does not prejudge the acceptance of the article, and the article as it stands already contributes a good deal of knowledge on the subject.

We agree with the reviewer that not including the internal work does not complete the mechanical analysis of running. It was a choice taken considering the paper is already lengthy. Please note, as suggested in the conclusion and perspectives section, that we are currently writing a follow-up paper which focuses on the inner and outer limb kinematic and muscular strategies used by the subject and the internal work should be included in this paper.

Considering that the calculation for internal work should not only look at the sagittal plane but also the frontal and transverse planes. The methodology for rotating the referential needed to be introduced. This was one of the goals of this paper.

Nevertheless, we specifically added the notion that internal work should be studied in the perspectives.

# I suggest that future studies check the parameters of the mass-spring system and the landing-takeoff asymmetries of the running. Some indirect evidence (such as Recovery and ta/tc relation) indicates that the elastic mechanism is disturbed in these curve conditions (not compulsory including in the paper, just a rationale).

Agreed. We have an ongoing project which is focused on adapting the spring-mass model presented by McMahon & Cheng, 1990 in three dimensions.

# Please move the Participants section to the beginning of the methods. Similarly, the selection of the analysed steps section can be between data collection and analysis.

We have moved the participants section to the beginning of the methods. However, as there are references to equations 4 and 6 in the selection of analysed steps, we find it should remain below the explanation.

Minor points

Ln 63 – is not

Done.

Ln 276-277 - I've probably missed this information, but please clarify how Wcom+ and Wcom- arrive at the final Wcom value (using only positive increments and normalizing by mass and stride or step length). I suggest using a current reference together to primary reference (e.g. Cavagna et al. 1963 or Cavagna et al. 1976) for this calculation.

We have added the following sentence:

The mass specific and step normalised positive (W_Y^+, W_X^+, W_z^+ and W_com^+) and negative (W_Y^-, W_X^-, W_z^- and W_com^-) works were normalised by the subject mass and step length [3,34,38].

 Cavagna et al., 1976

34. Peyre-Tartaruga et al., 2021

38. Mesquita et al., 2023

Ln 332 - ])

We have kept this notation following the recommendations of https://courses.lumenlearning.com/suny-osalgebratrig/chapter/domain-and-range/

Ln 372 – consider using exact p’s (P = 0.008 instead of P < 0.008) where possible.

Thank you we have checked this.

Ln 378 – occurs instead of seems to appear.

Done.

Ln 380 – as instead of than

Done.

Ln 438 – It is

Done.

Ln 445 – did not

Done.

Reviewer #2: General comments

This manuscript aims at investigating how the trajectory of the centre of mass of the body and the work done to maintain its movements relative to the surroundings are modified as a function of running speed and radius of curvature. The aim is commendable. Authors some curve running adaptations which provide valuable insights for sports sciences, training and performance in sports with multidirectional movements. Overall, the authors manage to fulfil their aim properly.

Specific comments: In which exact time of the foot contact does the referential change and why? How do you cope with some foot rotation during stance?

The rotation of the referential was done at the instant preceding the contact phase (1 ms) prior to contact as shown in Fig. 2. We have included this information in the manuscript which now reads:

At the last instant prior to the contact phase, the reference frame was rotated around the vertical z-axis so that the Y-axis of the new reference frame corresponded to the direction of the velocity vector vh at touch-down (TD), which was roughly the direction of vh during the previous aerial phase (Fig. 2B, left).

Considering the foot rotation: 

As the rotation was done the instant prior to foot contact, all movement occurring after this instant is considered in the reaction forces applied by the subject to turn their body over the foot. For the calculations of Wcom this does not affect our results.

Below, we verified the variation of foot angle (τ) in the horizontal plane while the foot was flat on the ground as a function of speed in both radii of curvature (see figure attached). τ was defined as the angle formed between the x-y position of the markers placed on the 5th metatarsal, heel and the horizontal. The variation was measured as the standard deviation of the τ angle between the point when the heel touches the ground until when the heel lifted off the ground by 0.05m, or so when the foot was flat on the ground. This was chosen as a means to verify that the foot was not skidding or rotating over the ground rather than measuring the ankle-roll over the toes during contact.

Results show that the variation of the foot angle is on average less than 2.5° over all traces. As mentioned in a reply to reviewer one, we are in the process of writing a follow-up paper which focuses on the inner and outer limb kinematic and muscular strategies used by the subject, segment positioning and differences will be explained in detail.

We have added this explanation in the supplementary materials with the figure and in the sentence explaining the rotation we changed this as follows:

As the foot is fixed on the ground during most of the contact phase (3) (

(line 181-2) “The contact phase (tc) corresponds to the period during which Fz > 10% of body weight (BW)…” Why? Isn’t it too much to detect real flight time (10% of 74.91 kgf is 7.491 kgf)?

We agree with the reviewer. The 10% of BW threshold was chosen to match a previous study conducted in conditions with a high signal to noise ratio (Mesquita et al., 2023). We have changed this from a percentage threshold to a fixed threshold at 30 N. This threshold is dependent on the noise of the instrumentation. Indeed, such a threshold should be dependent on the signal noise and not on the subject’s body weight (e.g., when observing ground reaction forces in children landing from a jump through various age groups, a body weight threshold would be rather variable (Toussaint et al., in preparation).

In the present case, when we compare a 10% of BW cut-off to a fixed 30 N cut-off for the contact time, the difference between both is of 3 m s. However, for the flight time this difference is considerable, an average difference of 10 m s.

All findings remain unchanged after this correction. We have fixed the text, the statical analyses and all relevant figures accordingly.

Minor comments

(line 66) … (Wcom) have not…

Done.

(l153) please, do not use acronyms in headings;

We have removed acronyms from all headings.

(l289) … mass: 74.91…

This was corrected.

(l441) sic?

This is Latin for as it appears in the original text. In order to make this clearer we have added quotation marks.

(l447) … circle (except…

Thank you.

---

## [Decision Letter · Decision Letter 1]

31 Jan 2024

Kinetics and mechanical work done to move the body centre of mass along a curve

PONE-D-23-40768R1

Dear Dr. Mesquita,

We’re pleased to inform you that your manuscript has been judged scientifically suitable for publication and will be formally accepted for publication once it meets all outstanding technical requirements.

Kind regards,

Yury Ivanenko

Academic Editor

PLOS ONE

Additional Editor Comments (optional):

Reviewers' comments:

Reviewer's Responses to Questions

**Comments to the Author**

1. If the authors have adequately addressed your comments raised in a previous round of review and you feel that this manuscript is now acceptable for publication, you may indicate that here to bypass the “Comments to the Author” section, enter your conflict of interest statement in the “Confidential to Editor” section, and submit your "Accept" recommendation.

Reviewer #2: All comments have been addressed

2. Is the manuscript technically sound, and do the data support the conclusions?

Reviewer #2: Yes

3. Has the statistical analysis been performed appropriately and rigorously? 

Reviewer #2: Yes

4. Have the authors made all data underlying the findings in their manuscript fully available?

Reviewer #2: Yes

5. Is the manuscript presented in an intelligible fashion and written in standard English?

Reviewer #2: Yes

6. Review Comments to the Author

Reviewer #2: General comments

I do not have any further particular concerns to express about the manuscript. Authors addressed sufficiently all points raised by the two reviewers.

7. PLOS authors have the option to publish the peer review history of their article (what does this mean?). If published, this will include your full peer review and any attached files.

Reviewer #2: No
